# Serum Cystatin-C is linked to increased prevalence of diabetes and higher risk of mortality in diverse middle-aged and older adults

Kevin A. González[1], Ariana M. Stickel[1], Sonya S. Kaur[2], Alberto R. Ramos[2], Hector M. González[1], Wassim Tarraf[3]*

1 Department of Neurosciences and Shiley-Marcos Alzheimer's Disease Research Center, University of California San Diego School of Medicine, San Diego, California, United States of America, 2 Department of Neurology, University of Miami Miller School of Medicine, Miami, Florida, United States of America, 3 Department of Healthcare Sciences and Institute of Gerontology, Wayne State University, Detroit, Michigan, United States of America

* wassim.tarraf@wayne.edu

**Data Availability Statement:** Data cannot be shared publicly because of you must comply with the Health and Retirement Study (HRS) data

## Abstract

### Objective

Type 2 Diabetes Mellitus (henceforth diabetes) affects roughly 35 million individuals in the US and is a major risk factor for cardiovascular and kidney disease. Serum Cystatin-C is used to monitor renal function and detect kidney damage. Recent research has focused on linking Cystatin-C to cardiovascular risk and disease, but most findings focus on small sample sizes and generalize poorly to diverse populations, thus limiting epidemiological inferences. The aim of this manuscript is to study the association between Cystatin-C, diabetes, and mortality and test for possible sex or racial/ethnic background modifications in these relationships.

### Methods

We analyzed 8-years of biennial panel data from Health and Retirement Study participants 50-years and older who self-identified as White (unweighted N (uN) = 5,595), Black (uN = 867), or Latino (uN = 565) for a total of uN = 7,027 individuals. We modeled diabetes and death over 8-years as function of baseline Cystatin-C (log transformed) adjusting for covariates and tested modifications in associations by race/ethnic background and sex.

### Results

Mean log Cystatin-C at visit 1 was 0.03±0.32 standard deviation. A 10% increase in Cystatin-C levels was associated with 13% increased relative risk of diabetes at baseline (11% and 9% by years 4 and 8). A 10% increase in Cystatin-C was highly associated with increased relative risk of death (28% and 31% by years 4 and 8). These associations were present even after adjusting for possible confounders and were not modified by sex or racial/ethnic background.

policies and agreement. Data are available from the HRS Institutional Data Access / Ethics Committee (create an account at https://hrsdata.isr.umich.edu/rda) for researchers who meet the criteria for access to confidential data. Links to the minimum datasets to perform this analysis can be found in S1 Data. If the researcher is provided access to both the RAND and biomarker files from 2006 to 2016, then this study can be fully replicated.

**Funding:** Dr. Hector González and colleagues are supported by R01-AG048642, RF1 AG054548 and RF1 AG061022 (National Institute on Aging). Dr. Hector González also receives additional support from P30AG062429 and P30AG059299 (National Institute on Aging). Additionally, Kevin González received support from the NSF GRFP. The funders had no role in study design, data collection and analysis, decision to publish, or preparation of the manuscript.

**Competing interests:** The authors have declared that no competing interests exist.

## Conclusion

Despite differential risks for diabetes and mortality by racial/ethnic groups, Cystatin-C was equally predictive of these outcomes across groups. Cystatin-C dysregulations could be used as a risk indicator for diabetes and as a warning sign for accelerated risk of mortality.

## Introduction

Thirty-five million U.S. adults had Type 2 Diabetes Mellitus (henceforth diabetes) in 2018, with 7.3 million of those cases being undiagnosed [1]. Diabetes is a risk factor for cancer [2], kidney disease [3], heart disease [4], stroke [5], and mortality [6]. Increased mortality in individuals with diabetes could be due to elevated risk of heart disease and stroke [7]. The total cost of diabetes care in the U.S. is expected to rise from 500 billion U.S. dollars in 2015 up to 714 billion dollars in 2030 [8]. The prevalence of diabetes varies by ethnic/racial background: Non-Latino Whites (henceforth Whites) have the lowest prevalence (11.9%), followed by Latinos (14.7%), Asians (14.9%), and Non-Latino Blacks (henceforth Blacks) (16.4%) [1]. Diabetes prevalence also varies by sex: men have higher prevalence of diabetes compared to women (14% for men vs 12% for women in the U.S) [1]. Most concerning, around 35% of US adults meet criteria for prediabetes based solely on HbA1c levels. Ethnic/racial minorities often face more barriers to accessing care which has implication to forgone or delayed diagnosis of diabetes. Latinos (20.0%) and Blacks (11.4%), for example, are more likely to be uninsured compared to Whites (7.8%) [9]. Latinos have the highest rate of undiagnosed diabetes [1], and their true prevalence may be greater than Blacks depending on Latino background. For example, estimates from the Hispanic Community Health Study/Study of Latinos indicated rates as high as 18.3% in a diverse Latino population of 18–74 years [10]. Despite higher rates of diabetes and largely worse socioeconomic profiles, Latinos have lower mortality rates (also known as Hispanic mortality paradox [11]), whereas the mortality rate for Blacks is higher than Whites [12]. Latinos with end stage renal disease, a common complication among those with diabetes, also show a survival advantage relative to their White peers, and a similar but less robust advantage is seen in Blacks [13].

Recent work, including a meta-analyses, show clear evidence for increased diabetes [14] and mortality [15] risk for those with elevated Cystatin-C. Some evidence suggests that this association may be modified by race, age, sex, as well as disease duration, with longer duration correlating with higher levels of this biomarker [14]. Cystatin-C is a protein regularly expressed by cells in the body [16]. The glomerulus filter this protein and it does not return to the bloodstream [16]. Thus, serum Cystatin-C can be used to measure glomerular filtration, with higher levels suggesting worse filtration (National Kidney Foundation). Cystatin-C may be a sensitive biomarker for early kidney disease detection [17], as well as a clinical flag to improve cardiovascular disease prediction and potentially decrease mortality [16]. While the connection between Cystatin-C and diabetes may be through early kidney disease detection, there may be other possible pathways. Cystatin-C has been linked to increased prevalence of metabolic syndrome, higher body mass index (BMI), waist circumference, and inflammation [18, 19], all of which have complex relationships with diabetes risk and mortality [20–22]. Thus, Cystatin-C dysregulations could be an important predictor and early marker for diabetes risk.

Recent research efforts have focused on linking Cystatin-C to onset and progression of cardiovascular and metabolic risk and disease, but most findings are based on small sample sizes

and generalize poorly to diverse populations, thus limiting appropriate epidemiological inferences [14, 23, 24]. Limited work has examined links between Cystatin-C, mortality and diabetes and tested sex and race/ethnic modifications [25–28]. Shlipak et al, 2006, used Health ABC data, from a cohort of Black and White participants ages 70–79 (N = 3,075) in 1997–1998, and found that each quantile increase in Cystatin-C was associated with increased mortality risk regardless of sex and background [25]. Likewise, Peralta et al, 2013 using data on Mexican Americans (ages 60–101, N = 1,435) from the Sacramento Area Latino Study of Aging (SALSA) study (years 1998–1998), found increased mortality risk in those with elevated Cystatin-C [26]. Sabanayagam et al (2013) used data from non-Hispanic Whites, non-Hispanic Blacks, and Mexican Americans from the National Health and Nutrition Examination Survey (ages 20+,N = 2,033; years 1999–2002) to asses prediabetes risk and reported that elevated Cystatin-C is associated with increased prediabetes not subject to modification by sex or background [27]. Lastly, Sahakyan et al (2011) using state level data from Wisconsin (ages 43–84, N = 4,936; years 1987–1988) reported significant associations between Cystatin-C and an increased risk of type 2 diabetes [28]. Our study expands on this work by using more recent large nationally representative data on adults middle aged and older (51+ years), with multiple time-point measurements for outcomes of interest, incorporating multiple operationalizations of the exposure (Cystatin-C, eGFR [estimated glomerular filtration rate]), and explicitly accounting for the risk of death directly in the outcome. Specifically, we: 1) examine associations between Cystatin-C levels and diabetes prevalence and mortality risk, and 2) explicitly test for race/ethnic background and sex modifications on Cystatin-C. We hypothesize that Cystatin-C increments will be associated with increased odds of diabetes as well as mortality. We also hypothesize that Blacks, Latinos, and men will have stronger (more risk) associations between Cystatin-C and outcomes given known variabilities in diabetes and renal function in these groups.

## Methods

### Data

We used data from the Health and Retirement Study (HRS), a nationally representative sample of US adults ages >50 years. The HRS is a longitudinal study that surveys around 20,000 individuals and is funded by the National Institute of Aging and the Social Security administration. The purpose of HRS is to provide multifaceted data to answer questions related to aging outcomes. The HRS collects data through biennial survey of cohorts started in 1992 and replenished every 6-years, on average, to account for mortality and sample attrition. Biomarker data collection was initiated in 2006, and is also collected biennially, on a rotating basis (half of the sample is targeted for data collection every other wave). The sampling framework, data collection strategies, survey and biological data modules descriptions, and detailed explanation of the HRS have been published elsewhere [29]. For this work, self-reported health, and sociocultural variables were obtained using the RAND HRS data [30]. Biomarker data was obtained from the HRS website, directly. HRS requires individuals to register and follow the procedures on their website to obtain the data. Links to data used can be found in S1 Data in S1 File. In line with previous work, we combined the measurements from the two half-samples participating in the biomarker study so that 8-years of follow up data would be available for each included individual (2006-2010-2014 and 2008-2012-2016 waves). Three waves of data were generated from these groupings: 1) visit 1: 2006 or 2008 data, 2) visit 2: 2010 or 2012 data, and 3) visit 3: 2014 or 2016 data (S1 Fig in S1 File).

## Ethics statement

The HRS study was approved by the UM Health Sciences/Behavioral Sciences IRB. Participants provided written informed consent before participating in the study as well as informed verbal consent before interviews. Further information on methods, design, and data availability can be found on the HRS website: https://hrs.isr.umich.edu/publications/biblio/9048.

## Outcomes

Our primary outcomes were diabetes and mortality. Diabetes classification was based on the American Diabetes Association's [31] criteria on Hemoglobin HbA1c thresholds: if an individual has HbA1c levels $\geq$ 6.5% (48 mmol/mol) or they self-reported as having diabetes, they were classified as having diabetes at the considered time point. We also assessed death status, as measured by HRS, at each visit. Further information on HRS biomarker data and mortality has been published elsewhere [32, 33]. For visit 1, we generated a dichotomous indicator to ascertain diabetes status (no diabetes, diabetes). We generated a trichotomous indicator for visits 2 and 3 representing the following groupings: 1) no diabetes, 2) diabetes, and 3) death status.

## Exposures

The main exposure was based on the serum Cystatin-C levels at visit 1 (2006 and 2008; please see above). Because Cystatin-C was not normally distributed, we used the natural logarithm to adjust the skew and allow to test linear associations with the outcomes of interest. We additionally generated estimated glomerular filtration rate values from Cystatin-C using the following equation described by Inker et al: $76.7 \times$ serum Cystatin-C $^{-1.19}$ [34]. This formula was used over the eGFR formula [35] developed by CDK-EPI, which includes corrections for sex and racial/ethnic background, because regression models include adjustments for sex and racial/ethnic background. Three categories were generated based on this indicator: 90+ (Normal kidney function), 60–89 (Mild kidney dysfunction), and 15–59 (Moderate kidney dysfunction). Those with eGFR <15 were excluded in primary analysis using eGFR. In sensitivity models, those with eGFR <15 were included in the moderate kidney dysfunction group.

## Analytical sample

Inclusion criteria were based on participation in the 2006 or 2008 waves (henceforth visit 1 of biomarker data collection) and provision of biomarker data to allow Cystatin-C measurement (n = 13,064). Additionally, given the scope of data collection in HRS, individuals had to be > 50 years of age (n = 12,725). We excluded (n = 264) individuals who did not identify as Black, Latino, or White, and who did not participate in visit 2 (n = 2,603) or visit 3 (n = 1,741) or had missingness on any of our covariates (n = 1,090). Our final analytic sample included unweighted n = 7,027 (the equivalent of 151,557,594 million middle aged and older adults when weighted). In analysis using eGFR, we also excluded participants with eGFR below 15 (n = 53) for a final analytic sample of n = 6,974. A flowchart of inclusion can be found in S2 Fig in S1 File.

## Covariates

Covariates were based on visit 1 data (2006 or 2008) and included: sex (male, female), continuous age, a trichotomous indicator for race/ethnic background (Non-Latino White, Black, Latino; excluded otherwise), an indicator for education (<12 years, 12 years, >12 years), a binary indicator for high-density lipoprotein (HDL) (low [<50mg/dL for women or <40mg/

dL for women] men, normal otherwise), a continuous measure of total cholesterol, a continuous measure of body mass index (BMI), a continuous measure of C-reactive protein (CRP), a binary indicator for alcohol consumption (not drinker, drinker), and a binary indicator for smoking status (not smoker, smoker). Covariates were identified based on previous work examining links between cystatin-C and diabetes (e.g. [36, 37]), and as covered by Luo et al's meta-analysis [15].

## Statistical analyses

All analyses were performed using Stata 16 SE. First, we generated descriptive statistics to characterize the target population overall and by race/ethnic background (Table 1). To test for heterogeneity of Cystatin-C across time points, we created mean and standard deviation tables for log Cystatin-C values based on diabetes and mortality (S1 Table in S1 File). The prevalence of diabetes and mortality across visits was included in S2 Table in S1 File. To test the associations between Cystatin-C and prevalent diabetes and mortality, we fit four multinomial logistic regression models for each of the three time points. At each time point, we fit 1) crude (unadjusted) model; 2) age, sex, race/ethnic background, and education adjusted models; 3) age, sex, race/ethnic background, education, BMI, drinking, and smoking adjusted model; and 4) fully adjusted (all covariates) models. From the above models, separate relative risk ratios for both diabetes and death with 95% confidence intervals were calculated (Table 2 for Cystatin-C; S3 Table in S1 File for eGFR). We plotted post-hoc marginal estimates derived from the fully adjusted models along with 95% confidence intervals (Figs 1 and 2).

Modifications by racial/ethnic background were subsequently assessed using multinomial logistic models including interactions between race/ethnic background and Cystatin-C. Results from post-hoc F-tests assessing significance of interactions are included in S4 Table in S1 File and estimates from interaction models are available in S5 Table in S1 File. Marginal estimates for racial/ethnic background modification are found on S3 and S4 Figs in S1 File. We also tested for sex modifications in Cystatin-C effects. First, we generated descriptive statistics based on sex groupings (S6 Table in S1 File). Second, we tested for modifications by including and interaction between sex and Cystatin-C in the multinomial logistic models and performed survey adjusted F-tests to obtain the statistical significance of the interactions (S7 Table in S1 File). Marginal estimates along with 95% confidence intervals were included in S5 and S6 Figs in S1 File. Lastly, we performed sensitivity analysis, repeating the specifications of the multinomial logit models detailed above, with categorical eGFR by including those with eGFR < 15 into the moderate dysfunction group (S8 Table in S1 File).

## Results

Summary statistics of the target population can be found in Table 1. Fifty three percent of participants were women and 85% were White. Average age was 66-years, mean BMI was 29 (kg/m2) and mean log Cystatin-C level was 0.03±0.32 standard deviation (SD) with a range of -1.46–2.31 at visit 1 (1.10±0.53 SD with range of 0.23–10.17 in the original scale). The prevalence of diabetes did not differ substantially over the considered time points (21% at visit 1, 22% at visit 2, and 21% at visit 3). However, mortality increased by 10–15% at each wave (15% at visit 2 and 25% at visit 3). We found significant differences in health outcomes based on race/ethnic background. Whites, on average, were the oldest but had the lowest BMI, CRP, and Cystatin-C levels of the three groups. Blacks had the highest BMI, CRP, and Cystatin-C values of the three groups. Latinos had the highest rates of diabetes at visit 1 (34%) and Blacks had the highest rate at visit 3 (33%). Blacks also had the highest morality rates at both visit 2 (17%) and visit 3 (28%). Descriptive measures by sex groupings are found in S6 Table in

**Table 1. Descriptive characteristics by background.**

| | White | Black | Latino | Total | P-Value |
|---|---|---|---|---|---|
| **Unweighted N** | 5595 | 867 | 565 | 7027 | |
| **Weighted %** | 84.0 | 9.0 | 7.0 | | |
| **Sex** | | | | | |
| Male | 47.39 | 40.32 | 45.58 | 46.66 | P = 0.001 |
| Female | 52.61 | 59.68 | 54.42 | 53.34 | |
| **Education** | | | | | |
| <12 years | 12.82 | 34.10 | 51.38 | 17.09 | P<0.001 |
| 12 years | 34.35 | 31.36 | 23.30 | 33.40 | |
| >12 years | 52.82 | 34.54 | 25.32 | 49.50 | |
| **Drinker** | | | | | |
| Not Drinker | 42.02 | 62.49 | 53.36 | 44.53 | P<0.001 |
| Drinker | 57.98 | 37.51 | 46.64 | 55.47 | |
| **Smoker** | | | | | |
| Not Smoker | 42.31 | 41.05 | 45.42 | 42.39 | P = 0.443 |
| Smoker | 57.69 | 58.95 | 54.58 | 57.61 | |
| **Diabetes V1** | | | | | |
| No Diabetes | 81.06 | 67.44 | 65.68 | 78.90 | P<0.001 |
| Diabetes | 18.94 | 32.56 | 34.32 | 21.10 | |
| **Diabetes and dead V2** | | | | | |
| No Diabetes | 65.63 | 50.71 | 54.53 | 63.62 | P<0.001 |
| Diabetes | 19.85 | 32.04 | 32.32 | 21.70 | |
| Dead | 14.52 | 17.26 | 13.15 | 14.68 | |
| **Diabetes and dead V3** | | | | | |
| No Diabetes | 55.45 | 39.32 | 46.16 | 53.45 | P<0.001 |
| Diabetes | 19.28 | 33.15 | 32.37 | 21.31 | |
| Dead | 25.27 | 27.52 | 21.46 | 25.24 | |
| **HDL** | | | | | |
| Low | 72.35 | 74.80 | 65.40 | 72.14 | P = 0.016 |
| Normal | 27.65 | 25.20 | 34.60 | 27.86 | |
| **eGFR Visit 1**[†] | | | | | |
| Normal | 32.04 | 34.22 | 39.57 | 32.70 | P = 0.005 |
| Mild | 44.44 | 38.49 | 44.36 | 43.92 | |
| Moderate | 23.51 | 27.30 | 16.06 | 23.38 | |
| **eGFR Visit 2**[†] | | | | | |
| Normal | 24.34 | 28.66 | 32.37 | 25.21 | P = 0.023 |
| Mild | 44.03 | 38.81 | 41.12 | 43.42 | |
| Moderate | 31.62 | 32.53 | 26.50 | 31.37 | |
| **eGFRVisit 3**[†] | | | | | |
| Normal | 22.01 | 27.46 | 23.12 | 22.54 | P = 0.091 |
| Mild | 44.23 | 37.15 | 47.54 | 43.85 | |
| Moderate | 33.75 | 35.39 | 29.33 | 33.60 | |
| **Mean (SD)** | | | | | |
| **Age** | 66.54 (9.57) | 64.38 (10.68) | 63.47 (9.86) | 66.16 (9.78) | P<0.001 |
| **BMI** | 28.29 (5.57) | 30.61 (8.63) | 29.41 (6.29) | 28.57 (5.93) | P<0.001 |
| **Cholesterol (mg/dL)** | 202.15 (38.83) | 203.16 (48.67) | 203.44 (48.09) | 202.32 (40.33) | P = 0.791 |
| **CRP (mg/L)** | 2.25 (3.97) | 3.41 (7.44) | 2.18 (2.85) | 2.35 (4.27) | P<0.001 |
| **Log Cystatin-C V1** | 0.03 (0.30) | 0.07 (0.49) | 0.00 (0.39) | 0.03 (0.32) | P = 0.065 |

(*Continued*)

**Table 1.** (Continued)

|  | White | Black | Latino | Total | P-Value |
|---|---|---|---|---|---|
| **Log Cystatin-C V2** | 0.08 (0.31) | 0.13 (0.51) | 0.03 (0.40) | 0.08 (0.33) | P = 0.034 |
| **Log Cystatin-C V3** | 0.11 (0.30) | 0.12 (0.47) | 0.08 (0.38) | 0.11 (0.32) | P = 0.586 |

BMI = Body Mass Index; HDL = High Density Lipoprotein; CRP = C-reactive protein. eGFR = Glomerular filtration rate. Log = natural logarithm

[†]: Normal = Normal kidney function; Mild = Mild kidney dysfunction; Moderate = Moderate kidney dysfunction

S1 File. Compared to men, women were older and less likely to report having education beyond high school (>12 years). Women had higher cholesterol and CRP levels, but lower rates of mortality and diabetes compared to men at all time points.

Differences in log Cystatin-C based on diabetes or death status are presented in S1 Table in S1 File. Log Cystatin-C values at visit 3 were higher for every group compared to visit 1, and trends were consistent for all visits: those without diabetes had the lowest Cystatin-C levels, followed by those with diabetes and those that died over the observation period. Lastly, we performed cross tabulations across the visits to test for spread among the groups. Those with diabetes at visit 1, while only consisting of 21% of the cohort, represented more than 30% of the dead at visit 2 and visit 3 (S2 Table in S1 File).

Results from the multinomial logit models can be found in Table 2, Fig 1, and S3 Table in S1 File. Higher log Cystatin-C values were associated with increased odds of diabetes and mortality at each time point. The following formula: $10 \cdot \log(\text{Odds Ratio})$ was used to convert odds ratios of log Cystatin-C to percentage risk increase in outcome based on 10% increase in Cystatin-C values (i.e. based on the original metric). A 10% increase in Cystatin-C, measured at visit 1, was associated with 13% increase in the odds ratios of diabetes (log Cystatin-C OR $_{\text{visit-1}}$ =

**Table 2. Association between log cystatin-C and diabetes and death status at visit 1, visit 2, and visit 3.**

|  | Visit 1 Diabetes | | | |
|---|---|---|---|---|
|  | **M0** | **M1** | **M2** | **M3** |
|  | **OR/CI** | **OR/CI** | **OR/CI** | **OR/CI** |
| **Log Cystatin C** | 3.69*** [3.02;4.51] | 3.43*** [2.70;4.37] | 2.46*** [1.93;3.13] | 2.35*** [1.83;3.02] |
|  | **Visit 2 Diabetes** | | | |
| **Log Cystatin C** | 3.00*** [2.33;3.86] | 2.57*** [1.97;3.35] | 1.56** [1.17;2.08] | 1.49* [1.10;2.02] |
|  | **Visit 2 Dead** | | | |
| **Log Cystatin C** | 16.23*** [12.04;21.89] | 5.72*** [4.24;7.72] | 4.90*** [3.57;6.73] | 4.28*** [3.06;6.00] |
|  | **Visit 3 Diabetes** | | | |
| **Log Cystatin C** | 2.45*** [1.86;3.24] | 2.38*** [1.77;3.19] | 1.42* [1.03;1.96] | 1.43* [1.02;2.00] |
|  | **Visit 3 Dead** | | | |
| **Log Cystatin C** | 22.68*** [17.07;30.14] | 7.37*** [5.43;10.00] | 6.01*** [4.41;8.19] | 5.26*** [3.75;7.39] |

OR = Odds Ratio; CI = Confidence Interval; Log = natural logarithm

M0: No adjustment

M1: Age, sex, background, and education.

M2: M1 + BMI, smoking status, and drinking status.

M3: M2 + HDL, total cholesterol, and C-reactive protein.

*: p<0.05.

**: p<0.01.

***:p<0.001

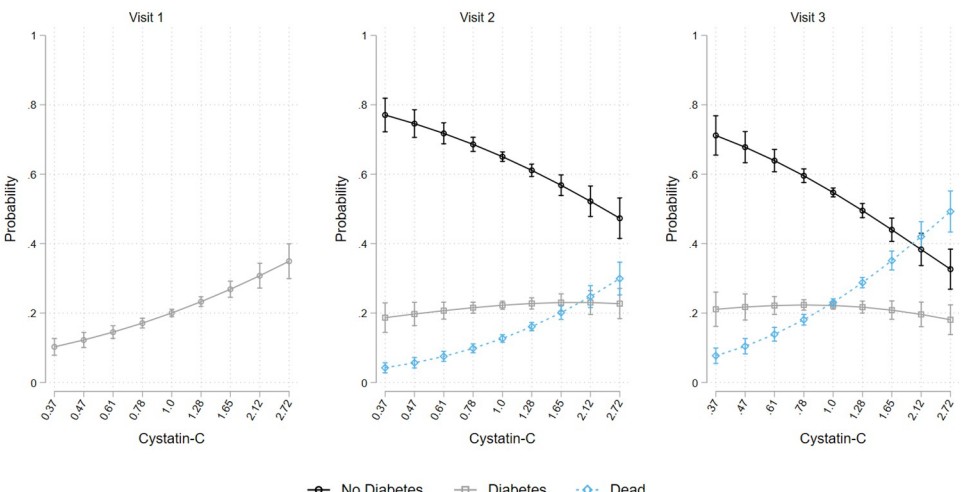

**Fig 1. Prevalence of (marginal probability and 95% confidence interval) no diabetes, diabetes, and death status at visit 1, visit 2, and visit 3 over Cystatin-C levels.**

3.69, p<0.001) at visit 1, 11% at visit 2 (OR visit-2 = 3.00, p<0.001), and 9% at visit 3 (OR visit-3 = 2.45, p<0.001). Covariate adjustment attenuated but did not fully explain these associations. We also found significant associations between elevated Cystatin-C and mortality. A 10% increase in Cystatin-C was associated with 28% increased mortality risk at visit 2 (OR visit-2 = 16.23, p<0.001) and 31.1% increased mortality at visit 3 (OR visit-3 = 22.68, p<0.001). Marginal probability estimates visually depicting these associations are included in Fig 1.

In eGFR models, moderate filtration loss (eGFR 15–90) was associated with increased odds ratios for diabetes at all time points (OR$_{visit-1}$ = 2.60, p<0.001; OR$_{visit-2}$ = 2.11, p<0.001; OR$_{visit-3}$ = 1.90, p<0.001). Mild filtration loss was associated with increased odds ratios for diabetes at visit 2 and visit 3 (OR$_{visit-2}$ = 1.26, p<0.05; OR $_{visit-2}$ = 1.22, p<0.05) but the associations were explained through covariates adjustment. Both moderate (OR$_{visit-2}$ = 8.51, p<0.001; OR$_{visit-3}$ = 9.43, p<0.001) and mild (OR$_{visit-2}$ = 1.96, p<0.001; OR$_{visit-3}$ = 2.12, p<0.001) filtration loss were associated with increased odds ratios of mortality at visit 2 and visit 3, and consistently so after full covariates adjustment (S3 Table in S1 File and Fig 2).

We did not find consistent evidence for race/ethnic modifications in associations between Cystatin-C and diabetes or mortality (S4 Table in S1 File). Higher levels of Cystatin-C among Blacks significantly lowered the odds ratios for mortality (OR$_{visit-2}$ = 0.22, p<0.001; OR$_{visit-3}$ = 0.33, p<0.05) and diabetes (OR $_{visit-2}$ = 0.46, p<0.05) compared to the reference White group, but these differences were explained by adjustments to covariates (S5 Table in S1 File; also see S3 and S4 Figs in S1 File for eGFR). The tested interactions between Cystatin-C levels and sex were not statistically significant and consistently so in crude and adjusted models suggesting no evidence for differences in associations between Cystatin-C levels and either mortality or diabetes across men and women (S7 Table in S1 File; also see S5 and S6 Figs in S1 File for eGFR).

Results from sensitivity models are found in S8 Table in S1 File. The associations between moderate eGFR and both diabetes and mortality were stronger (accentuated) when we included individuals with eGFR < 15 into the moderate dysfunction group. However, there were no notable substantive differences between these estimates and those reported in S3 Table in S1 File (as described above).

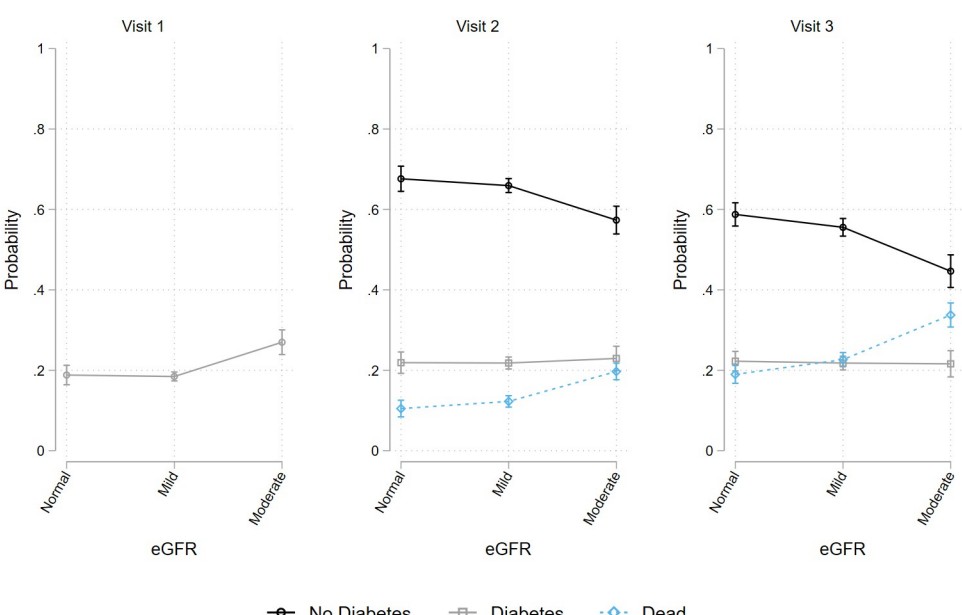

**Fig 2. Prevalence of (marginal probability and 95% confidence interval) no diabetes, diabetes, and death status at visit 1, visit 2, and visit 3 by eGFR.**

## Discussion

Using data from the HRS, a longitudinal study representative of middle aged and older adults (>50-years) from diverse race/ethnic groups in the US, we examined the associations between Cystatin-C and glomerular filtration rate, diabetes prevalence, and mortality. Three main findings emerged: 1) Higher Cystatin-C was associated with increased risk both diabetes and mortality and these associations were consistent for filtration loss, particularly for moderate levels of eGFR; 2) we found notable differences in Cystatin-C levels, diabetes prevalence, and mortality rates between Latinos, Blacks, and Whites, however, Cystatin-C was not differentially associated with either diabetes or mortality across the three groups after adjustment for covariables; 3) Lastly, Cystatin-C was similarly linked to diabetes and mortality in men and women.

Cystatin-C was associated with higher risk of diabetes and mortality at baseline and at each of the follow-up assessment periods. Furthermore, Cystatin-C was equally sensitive regardless of ethnic/racial background, despite notable differences in both exposure levels and outcomes across the considered groups. Our findings complement and expand evidence linking Cystatin-C, diabetes and mortality (treated independently) in other cohort studies, with different age ranges, race/ethnic background inclusion, and population representation [25–28]. We also expand on evidence published in recent meta-analyses [14] that had an overall sample size of n = 1,196 (n = 723 with diabetes and n = 473 controls) and relatively limited power. While this meta-analysis reported sex and background differences in Cystatin-C; our study suggests that these are likely baseline differences that might be explained by accounting for mortality, and do not modify the relationship between Cystatin-C and diabetes. Our mortality findings also complement a 2015 meta-analysis by Luo et al [15] and suggest that Cystatin-C control can potentially offer a targetable pathway for lowering diabetes risk and improving longevity.

There are several possible pathways by which elevated Cystatin-C could increase the risk of diabetes, and mortality. One possible pathway is through insulin resistance. Elevated insulin

resistance is associated with worse glomeruli filtration [38], and as such Cystatin-C could be a proxy for insulin resistance [39, 40]. Another possible pathway is through BMI, as higher BMI is linked to increased Cystatin-C levels [41]. Reuters et al, found that Cystatin-C was only associated with elevated diabetes risk in individuals with insulin resistance or central adiposity [42]. A 2019 review indicates that adipocytes, which are cells that store fat, interact with the nervous system and could be involved with glucose homeostasis [43]. Inflammation too, could play a role and several pathways through which inflammation (micro and extracellular) elevates diabetes risk and development have been explored in literature [44].

Blacks had higher rates of diabetes, mortality and higher Cystatin-C compared to Whites. The REGARDS study found lower prevalence rates of kidney dysfunction in Blacks compared to Whites; however, the relationship reversed as severity of dysfunction increased [45]. More recent research has confirmed faster progression towards end stage renal disease in Blacks [46, 47]. Paradoxically, Blacks with end stage renal disease have better survival rates compared to Whites [48]. A possible explanation for this could be survivorship bias: Blacks with chronic kidney disease had higher mortality compared to Whites if they were younger than 65 years old, but not if they were 65 years or older [49]. Our results are consistent with these mixed findings as we did not find group differences in mortality through Cystatin-C.

Latinos had higher rates of diabetes and lower rates of mortality compared to Whites. The higher rates of diabetes in Latinos have been documented extensively in literature and this evidence is in line with our results [50]. Latinos also have higher rates of end stage renal disease compared to Whites, although this difference is not completely explained by differences in diabetes prevalence [51]. Latinos, despite having worse cardiovascular health outcomes, have lower mortality compared to Whites (also called the Hispanic Mortality Paradox [11]). There are many factors (both protective and detractive) that affect and complicate health outcomes in Latinos (e.g. acculturation [52], and access to and experience with healthcare [53]). Our results suggest Cystatin-C is equally predictive of diabetes in Whites, Blacks, and Latinos after controlling for relevant covariates, and thus could be an important tool for healthcare providers to use across patient populations.

Despite lack of evidence for sex modifications in our work, there are important sex differences in kidney dysfunction between men and women. For example, a systematic review by Nitsch et al [54], points to greater all-cause mortality odds among men compared to women at higher glomerular filtration rates, but suggests that the relationship reverses at low filtration rates. A possible explanation for the lack of modification is that women with diabetes have higher mortality risks compared to men with diabetes [55]. The biological and sociocultural pathways to disease and mortality in women and men are complex and potentially require longer observation periods, a wider age range (starting in earlier middle age), and more specific stratification over population groups (e.g. for examining sex differences within and across race/ethnic groups). These complexities are beyond the limited scope of this work but should be further investigated.

## Strengths and limitations

Our findings should be interpreted in the context of a few limitations. First, sample sizes for Latinos and Blacks were small and might not represent the true diversity of these groups (e.g. differences across Latino heritage groups). Second, we only used Cystatin-C as measured at visit 1, which could affect reliability of results at visit 2 and 3. Third, our study did not account for age as a possible modifier for Cystatin-C. We also did not use longitudinal techniques to model repeated measures. Longitudinal generalized mixed models would not be appropriate in our case given that death is included as a competing risk (category) in our outcomes. As

such a different operationalization of the outcome would be required to model growth risk for diabetes independently. Furthermore, mixed modeling for complex survey sampling like the HRS is further complicated by the limited capabilities and required assumptions to account for the sampling design including stratification, clustering and appropriate weighting. Additionally modeling death jointly, require more sophisticated techniques, such as joint growth and discrete-time survival which go well beyond the scope of this study. We believe that our methodology is valid, represents a solid first step to expand the literature, and have restricted our inferences accordingly in text and discussion. The use of longitudinal models should be considered in future work to establish the robustness of our finding and inferences. Lastly, we had a limited availability of biomarkers, which minimizes our ability to adjust for possible confounders. Despite those limitations, our study has several strengths. First, we modeled repeated measurements of the outcome, which helps validate our results over time, and incorporated death as a competing risk. Second, we accounted for undiagnosed diabetes by using HbA1c, in addition to self-report, to assess diabetes status. This is critical since it addresses a gap in work that misses undiagnosed diabetes between race/ethnic groups. Lastly, our findings address a gap in research on Cystatin-C and diabetes, particularly regarding potential differential influences by race/ethnic background and sex.

A better understanding of how cystatin-C influences diabetes prevalence, and potentially its onset, can have significant health and healthcare implications. Interventions that target modifiable lifestyle, behavioral, and biological risks could be effective at lowering the risk for diabetes [56] and potentially improve downstream cardiovascular complications and lower the risk of death. Compared to previous work, our study offers important strengths.

## Conclusion

Despite differential risks for diabetes and mortality by racial/ethnic groups, Cystatin-C was equally predictive of these outcomes across groups. Cystatin-C, in addition to other biomarkers, could be used to create a comprehensive diabetes risk profile. In addition to its current use for monitoring renal health, Cystatin-C could be used as a risk indicator for diabetes as well as an early predictor of potentially premature mortality.

## Supporting information

**S1 File.**
(DOCX)

## Author Contributions

**Conceptualization:** Kevin A. González, Wassim Tarraf.

**Formal analysis:** Kevin A. González.

**Funding acquisition:** Hector M. González.

**Resources:** Ariana M. Stickel, Hector M. González, Wassim Tarraf.

**Supervision:** Ariana M. Stickel, Wassim Tarraf.

**Validation:** Sonya S. Kaur, Alberto R. Ramos, Wassim Tarraf.

**Visualization:** Kevin A. González.

**Writing – original draft:** Kevin A. González.

**Writing – review & editing:** Kevin A. González, Ariana M. Stickel, Sonya S. Kaur, Alberto R. Ramos, Hector M. González, Wassim Tarraf.

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
