## [Decision Letter · Decision Letter 0]

17 Jan 2022

PONE-D-21-38060Serum Cystatin-C is linked to increased prevalence of diabetes and higher risk of mortality in diverse middle-aged and older adultsPLOS ONE

Dear Dr. Tarraf,

Thank you for submitting your manuscript to PLOS ONE. After careful consideration, we feel that it has merit but does not fully meet PLOS ONE’s publication criteria as it currently stands. Therefore, we invite you to submit a revised version of the manuscript that addresses the points raised during the review process.

We look forward to receiving your revised manuscript.

Kind regards,

Aloysious Dominic Aravinthan, MBBS, FRCP, PhD

Academic Editor

PLOS ONE

Journal Requirements:

2. Please ensure that you have specified (1) whether consent was informed, (2) what type you obtained (for instance, written or verbal, and if verbal, how it was documented and witnessed). If your study included minors, state whether you obtained consent from parents or guardians. If the need for consent was waived by the ethics committee and (3) If you are reporting a retrospective study of medical records or archived samples, please ensure that you have discussed whether all data were fully anonymized before you accessed them and/or whether the IRB or ethics committee waived the requirement for informed consent. If patients provided informed written consent to have data from their medical records used in research, please include this information.

"Dr. Hector González and colleagues are supported by R01-AG048642, RF1 AG054548 and RF1 AG061022 (National Institute of Aging). Dr. Hector González also receives additional support from P30AG062429 and P30AG059299. Additionally, Kevin González received support from the NSF GRFP." 

Additional Editor Comments:

The strength of this study is the size of the cohort; however, as the reviewers point out there are a number of deficiencies in how it is presented and needs authors' attention.

Reviewers' comments:

Reviewer's Responses to Questions

**Comments to the Author**

1. Is the manuscript technically sound, and do the data support the conclusions?

Reviewer #1: Yes

Reviewer #2: Yes

2. Has the statistical analysis been performed appropriately and rigorously? 

Reviewer #1: Yes

Reviewer #2: Yes

3. Have the authors made all data underlying the findings in their manuscript fully available?

Reviewer #1: Yes

Reviewer #2: Yes

4. Is the manuscript presented in an intelligible fashion and written in standard English?

Reviewer #1: Yes

Reviewer #2: No

5. Review Comments to the Author

Reviewer #1: Excellent job in analyzing this large data, this would be considered a well done manuscript for future references. I do have few points as below:

1. Line 51: No mention of diabetes as a risk factor of cardiovascular disease and death, while knowing that cardiovascular disease is the most common cause of death in patients with diabetes, hence relevant to noticed increased relative risk of death among patients with elevated Cystatin-C.

2. Line 59: Hba1c better to written (henceforth) as HbA1c or HbA1c.

3. Authors could spend more time discussing and possibly explaining the reason of elevated Cystatin-C and increased risk of diabetes and death in terms of pathophysiology.

Reviewer #2: Whilst the study has an advantage of a relatively large sample size and longitudinal design, it lacks originality and novelty. An extensive literature on the associations of Cystatin C with mortality and DM risk including various ethnic groups already exists (examples of some of those studies not referenced in the manuscript:

Shlipak MG, Wassel Fyr CL, Chertow GM, Harris TB, Kritchevsky SB, Tylavsky FA, Satterfield S, Cummings SR, Newman AB, Fried LF. Cystatin C and mortality risk in the elderly: the health, aging, and body composition study. J Am Soc Nephrol. 2006 Jan;

Peralta CA, Lee A, Odden MC, Lopez L, Zeki Al Hazzouri A, Neuhaus J, Haan MN. Association between chronic kidney disease detected using creatinine and cystatin C and death and cardiovascular events in elderly Mexican Americans: the Sacramento Area Latino Study on Aging. J Am Geriatr Soc. 2013

Sabanayagam C et all. Serum cystatin C and prediabetes in non-obese US adults. Eur J Epidemiol. 2013 Apr;

K. Sahakyan et all. Serum Cystatin C and the Incidence of Type 2 Diabetes Mellitus, Diabetologia 2011)

The manuscript does not read well in sections, particularly the results section, which needs to be presented in a clearer manner. Also, I would suggest placing the tables with the description of the findings within Results Section rather than Statistical Analyses. There are a number of small errors/typos throughout this this manuscript which needs correcting such as:

Line 132- error in eGRF categories

HDL categories- discrepancy in description on text/table 1

Table 1- units of measure in the table

Line 240 - unclear what authors mean by “lower association”

Duplication of reference: 11=13

Line 257 - Cystatin-C was associated with higher odds of diabetes and ??mortality at baseline

Please justify the choice of covariates on statistical grounds (unless biologically plausible mechanism linking covariate to the dependant variable exists) and provide the cut-off points for the categorical variables where missing.

The authors might want to take advantage of the availability of the Cystatin C data at three time points and apply more complex statistics such as repeated measures GLM to allow more robust longitudinal analysis.

6. PLOS authors have the option to publish the peer review history of their article (what does this mean?). If published, this will include your full peer review and any attached files.

Reviewer #1: No

Reviewer #2: No

---

## [Author Response · Author response to Decision Letter 0]

30 Mar 2022

Reviewer #1: Excellent job in analyzing this large data, this would be considered a well done manuscript for future references. I do have few points as below:

1. Line 51: No mention of diabetes as a risk factor of cardiovascular disease and death, while knowing that cardiovascular disease is the most common cause of death in patients with diabetes, hence relevant to noticed increased relative risk of death among patients with elevated Cystatin-C.

We thank the reviewer for pointing us to this omission and have now addressed this in the introduction (lines 90-96).

2. Line 59: Hba1c better to written (henceforth) as HbA1c or HbA1c.

We have updated the manuscript to consistently use HbA1c throughout the manuscript.

3. Authors could spend more time discussing and possibly explaining the reason of elevated Cystatin-C and increased risk of diabetes and death in terms of pathophysiology.

Thank you for your comment. We have included language to discuss pathophysiology (lines 298-307).

Reviewer #2: Whilst the study has an advantage of a relatively large sample size and longitudinal design, it lacks originality and novelty. An extensive literature on the associations of Cystatin C with mortality and DM risk including various ethnic groups already exists (examples of some of those studies not referenced in the manuscript:

Shlipak MG, Wassel Fyr CL, Chertow GM, Harris TB, Kritchevsky SB, Tylavsky FA, Satterfield S, Cummings SR, Newman AB, Fried LF. Cystatin C and mortality risk in the elderly: the health, aging, and body composition study. J Am Soc Nephrol. 2006 Jan;

Peralta CA, Lee A, Odden MC, Lopez L, Zeki Al Hazzouri A, Neuhaus J, Haan MN. Association between chronic kidney disease detected using creatinine and cystatin C and death and cardiovascular events in elderly Mexican Americans: the Sacramento Area Latino Study on Aging. J Am Geriatr Soc. 2013

Sabanayagam C et all. Serum cystatin C and prediabetes in non-obese US adults. Eur J Epidemiol. 2013 Apr;

K. Sahakyan et all. Serum Cystatin C and the Incidence of Type 2 Diabetes Mellitus, Diabetologia 2011)

Thank you, we now reference these manuscripts in the introduction section (90-105) and distinguish our manuscript.

The manuscript does not read well in sections, particularly the results section, which needs to be presented in a clearer manner. Also, I would suggest placing the tables with the description of the findings within Results Section rather than Statistical Analyses. There are a number of small errors/typos throughout this this manuscript which needs correcting such as:

Thank you we have revised for better clarity. In addition, we have conducted a thorough copy edit to fix aberrant errors and typos.

Line 132- error in eGRF categories – fixed (now lines 150-51)

HDL categories- discrepancy in description on text/table 1 - fixed

Table 1- units of measure in the table

Line 240 - unclear what authors mean by “lower association”. Updated text in ms

Duplication of reference: 11=13 - fixed

Please justify the choice of covariates on statistical grounds (unless biologically plausible mechanism linking covariate to the dependant variable exists) and provide the cut-off points for the categorical variables where missing.

Thank you. We now include a rationale for choice of covariates. Additionally, we included cutoff for categorical variables.

The authors might want to take advantage of the availability of the Cystatin C data at three time points and apply more complex statistics such as repeated measures GLM to allow more robust longitudinal analysis.

Unfortunately, repeated measures GLM, or longitudinal generalized mixed models would not be appropriate in our case given that death is included as a competing risk (category) in our outcomes. Mixed modeling for complex survey sampling like the HRS is also further complicated by the limited capabilities to account for the sampling design including stratification, clustering and appropriate weighting (particularly weighting the different levels). Accounting for death independently, would require more sophisticated techniques, such as joint growth and discrete-time survival which go well beyond the scope of this analyses. We believe that our methodology is valid and we restrict our inferences accordingly in text and discussion. We now add this rationale to the limitations section and point to the need for use of longitudinal models as future considerations. (Lines 343-354)

---

## [Decision Letter · Decision Letter 1]

8 Jun 2022

Serum Cystatin-C is linked to increased prevalence of diabetes and higher risk of mortality in diverse middle-aged and older adults

PONE-D-21-38060R1

Dear Dr. Tarraf,

We’re pleased to inform you that your manuscript has been judged scientifically suitable for publication and will be formally accepted for publication once it meets all outstanding technical requirements.

Kind regards,

Aloysious D Aravinthan, MBBS, FRCP, PhD

Academic Editor

PLOS ONE

Additional Editor Comments (optional):

Authors have addressed the reviewers' comments to my satisfaction.

Reviewers' comments:

Reviewer's Responses to Questions

**Comments to the Author**

1. If the authors have adequately addressed your comments raised in a previous round of review and you feel that this manuscript is now acceptable for publication, you may indicate that here to bypass the “Comments to the Author” section, enter your conflict of interest statement in the “Confidential to Editor” section, and submit your "Accept" recommendation.

Reviewer #1: All comments have been addressed

Reviewer #2: (No Response)

2. Is the manuscript technically sound, and do the data support the conclusions?

Reviewer #1: Yes

Reviewer #2: Yes

3. Has the statistical analysis been performed appropriately and rigorously? 

Reviewer #1: Yes

Reviewer #2: Yes

4. Have the authors made all data underlying the findings in their manuscript fully available?

Reviewer #1: Yes

Reviewer #2: Yes

5. Is the manuscript presented in an intelligible fashion and written in standard English?

Reviewer #1: Yes

Reviewer #2: Yes

6. Review Comments to the Author

Reviewer #1: Excellent job in analyzing this large data, this would be considered a well done manuscript for future references

Reviewer #2: (No Response)

7. PLOS authors have the option to publish the peer review history of their article (what does this mean?). If published, this will include your full peer review and any attached files.

Reviewer #1: No

Reviewer #2: No

---

## [Editor Report · Acceptance letter]

2 Sep 2022

PONE-D-21-38060R1 

Serum Cystatin-C is linked to increased prevalence of diabetes and higher risk of mortality in diverse middle-aged and older adults 

Dear Dr. Tarraf:

I'm pleased to inform you that your manuscript has been deemed suitable for publication in PLOS ONE. Congratulations! Your manuscript is now with our production department. 

Kind regards, 

on behalf of

Dr. Aloysious Dominic Aravinthan 

Academic Editor

PLOS ONE